# Viral Myocarditis—From Pathophysiology to Treatment

**DOI:** 10.3390/jcm10225240

**Published:** 2021-11-11

**Authors:** Heinz-Peter Schultheiss, Christian Baumeier, Ganna Aleshcheva, C.-Thomas Bock, Felicitas Escher

**Affiliations:** 1Institute of Cardiac Diagnostics and Therapy, IKDT GmbH, 12203 Berlin, Germany; christian.baumeier@ikdt.de (C.B.); ganna.aleshcheva@ikdt.de (G.A.); felicitas.escher@charite.de (F.E.); 2Division of Viral Gastroenteritis and Hepatitis Pathogens and Enteroviruses, Department of Infectious Diseases, Robert Koch Institute, 13353 Berlin, Germany; BockC@rki.de; 3Institute of Tropical Medicine, University of Tuebingen, 72074 Tuebingen, Germany; 4Department of Internal Medicine and Cardiology, Campus Virchow-Klinikum, Charité-Universitaetsmedizin Berlin, Corporate Member of Freie Universitaet Berlin and Humboldt-Universitaet zu Berlin, 13353 Berlin, Germany; 5DZHK (German Centre for Cardiovascular Research), Partner Site Berlin, Berlin, Germany

**Keywords:** viral myocarditis, autoimmunity, dilated cardiomyopathy, pathophysiology, endomyocardial biopsy, therapeutic approach

## Abstract

The diagnosis of acute and chronic myocarditis remains a challenge for clinicians. Characterization of this disease has been hampered by its diverse etiologies and heterogeneous clinical presentations. Most cases of myocarditis are caused by infectious agents. Despite successful research in the last few years, the pathophysiology of viral myocarditis and its sequelae leading to severe heart failure with a poor prognosis is not fully understood and represents a significant public health issue globally. Most likely, at a certain point, besides viral persistence, several etiological types merge into a common pathogenic autoimmune process leading to chronic inflammation and tissue remodeling, ultimately resulting in the clinical phenotype of dilated cardiomyopathy. Understanding the underlying molecular mechanisms is necessary to assess the prognosis of patients and is fundamental to appropriate specific and personalized therapeutic strategies. To reach this clinical prerequisite, there is the need for advanced diagnostic tools, including an endomyocardial biopsy and guidelines to optimize the management of this disease. The severe acute respiratory syndrome-coronavirus-2 (SARS-CoV-2) has currently led to the worst pandemic in a century and has awakened a special sensitivity throughout the world to viral infections. This work aims to summarize the pathophysiology of viral myocarditis, advanced diagnostic methods and the current state of treatment options.

## 1. Introduction

The term “myocarditis” was introduced by Jean-Nicolas Corvisart in the early 19th century. The exact incidence of myocarditis (acute and chronic) remains unclear, mainly due to the heterogeneity in symptomatology ranging from nonspecific symptoms of fatigue to fulminant acute heart failure with the need for heart transplantation [1].

Most cases of myocarditis are probably caused by infectious agents, although often, the pathogen cannot be detected after the onset of the disease. Different virus genomes have been found to be associated with myocarditis: especially a high prevalence of parvovirus B19 (B19V), member of the *Parvoviridae* family genus *Erythroparvovirus* and human herpesvirus 6 (HHV6), indicating a shift in the last years [2,3,4,5] (Table 1).

Viral persistence in the myocardium is associated with progressive deterioration of left ventricular ejection fraction (LVEF), whereas elimination of the viral genomes led to a marked improvement in left ventricular (LV) function [43,44]. The presence of active virus replication, which has received less attention so far, probably plays a decisive role in this process [45,46,47]. 

Pathogens, including viruses, can secondarily trigger autoimmune mechanisms. Most likely, at a certain point, several etiological types merge into a common pathogenic autoimmune process, leading to chronic inflammation and tissue remodeling, ultimately resulting in the clinical phenotype of dilated cardiomyopathy (DCM) [48,49,50,51,52].

Acute myocarditis is diagnosed by histological, immunological and immunohistochemical criteria and clinically implies a short time elapsed from the onset of symptoms and diagnosis, while dilated inflammatory cardiomyopathy (DCMi) indicates myocarditis in association with cardiac dysfunction. Chronic myocarditis could represent an intermediate stage with a longer duration of symptoms (>1 month) in patients with persisting myocardial inflammation.

Endomyocardial biopsy (EMB) with histology, immunostaining for inflammation and polymerase chain reaction (PCR) remains the gold standard for the diagnosis due to its definitive capacity and etiologic diagnosis (viral or immune-mediated) in myocarditis and DCMi [1,53,54,55]. Here, noninvasive diagnostics fail when infectious agents are involved because they cannot detect or quantify different viral types or subtypes or the degree and quality of inflammation to identify specific forms of immune response. Moreover, the use of advanced diagnostic tools has led to better identification of the etiology of myocarditis and to a new interest in the mechanisms of the inflammatory process in the heart. Understanding the underlying molecular mechanisms is necessary to assess the prognosis of patients and is fundamental to appropriate specific and personalized therapeutic strategies.

The severe acute respiratory syndrome-coronavirus-2 (SARS-CoV-2) infection and the resulting clinical manifestation of coronavirus disease 2019 (COVID-19) spread rapidly throughout the world. Cardiac involvement in the current COVID-19 epidemic is reported; however, whether myocardial injury may be related to the effects of the generalized infection or direct cardiac involvement is still under investigation. Nevertheless, this is the worst pandemic in a century and has awakened a special sensitivity to viral infections throughout the world.

Although myocarditis and DCMi are associated with high morbidity and mortality, worldwide recommendations for diagnosis and treatment are vague due to a lack of data from large, randomized studies. The low awareness of this disease leads to an even larger knowledge gap that urgently needs to be filled. 

With the intention of summarizing the currently available knowledge on the pathophysiological mechanisms of viral cardiomyopathy, this review aims to shed light on viral etiology, advanced diagnostics and the current state of treatment options. 

## 2. General Pathophysiological Aspects 

### 2.1. Phases of Viral-Mediated Myocarditis

It stands to reason that the ability to predict, with reasonable accuracy, the clinical course and progression of disease in patients with viral myocarditis would facilitate resource management and the early application of certain therapeutic options, including pharmacological treatment [5,49,56,57]. To better understand the course of the disease and to make a prognostic statement, it is necessary to address the underlying pathophysiological process. After the acute phase of viral-mediated myocarditis, there are three commonly accepted clinical possibilities: (1) the virus is cleared without residual inflammation, resulting in complete healing; (2) the viral infection persists with or without inflammation; or (3) the viral infection results in autoimmune-mediated inflammation that persists despite clearance of the virus [4,52,58,59]. 

If the infectious agent is rapidly eliminated and the inflammatory process is finalized, the disease will heal with only minor changes in the myocardium. At this stage, the true causes of the disease can no longer be determined. These patients usually recover completely within weeks to months.

In contrast, when the viral infection has been overcome, and the antiviral immune response has subsided, but irreversible myocardial damage has already developed, the clinical picture evolves into DCM. In this situation, diagnostics have started too late, and they cannot elucidate the original causes of the disease. Therefore, current data argue for the need to identify patients at an early and still reversible stage of virus-associated heart disease [60] (Figure 1). The aim of any diagnostics is to enable a precise diagnosis that can differentiate between virus positivity or virus exclusion, proof of inflammation, characterization and intensity. This is possible by the use of the EMB to enable a specific, personalized and causal treatment option. 

### 2.2. Autoimmunity in Viral Myocarditis

The chronic immune stimulation or autoimmunity in chronic viral myocarditis is the result of incompletely overcoming viral infection or a response to the preceding virus or immune-mediated chronic tissue injury. By understanding this pathophysiology, it is obvious that viral diagnostics in EMB is always required when inflammation is confirmed. Both the persistent antigenic trigger by continuously synthesized viral proteins and the release of intracellular proteins from necrotic or apoptotic myocardial cells can stimulate chronic inflammation that may eventually involve the entire myocardium.

Autoimmune reactions—possibly favored by molecular mimicry—activate virus-specific T cells that attack the myocardium. High concentrations of cytokines (e.g., tumor necrosis factor (TNF), interleukin (IL)-1a, IL-1b, IL-2 and interferon- (IFN-) γ) are produced during this phase. These cytokines, together with antibodies against viral and cardiac proteins, further exacerbate the damage to the heart and impairment of systolic function due to disturbance of the contractile apparatus and matrix proteins [61,62,63,64,65]. 

Recent studies have investigated the T helper (Th) 17 effector functions during viral infections, including its critical role in the production and induction of proinflammatory cytokines and in the recruitment and activation of other immune cells. Thus, Th17 is involved in the induction of both pathogenicity and immunoprotective mechanisms observed in the host immune response to viruses and can also modulate immune responses [48].

Genetics may play a role in predisposing certain populations to be more susceptible to infection and possibly, to develop myocarditis as an autoimmune response once infected with a cardiotropic virus. The geographic and temporal distribution of viral strains identified in the past suggests that viral strains specifically affect local populations and evolve over distance, with a seasonal distribution likely [66].

## 3. Clinical Presentations and Diagnostics

Myocarditis is challenging to diagnose due to the heterogeneity of clinical presentations, which ranges broadly from asymptomatic to fulminant heart failure. The natural course of the disease seems unpredictable so far. There is no virus-specific phenotype of myocarditis or DCMi [4]. A viral infection affecting the respiratory or gastrointestinal tract and accompanied by systemic symptoms may precede the onset of cardiac symptoms, although the incidence of such a viral syndrome varies widely.

Viral serology is of limited use for diagnosis, especially in chronic myocarditis or inflammatory cardiomyopathy. Furthermore, IgG antibodies against cardiotropic viruses can be found in the blood of the general population without cardiac involvement [67].

Echocardiography is a valuable tool for detecting global or regional wall motion abnormalities, and myocardial strain patterns are of particular value for early diagnosis. Magnetic resonance imaging (MRI) with T1 and T2 sequencing and late gadolinium enhancement can be invaluable for visualizing structural changes, infiltration, inflammation, fibrosis and scarring [68]. However, one substantial criticism of a recent trial, which assessed the performance of different sequences on cardiac MRI to detect acute myocarditis compared with EMB performed in both ventricles, was the relatively low pre-test probability of a diagnosis of acute myocarditis [69]. Although imaging techniques, such as MRI, provide noninvasive tissue characterization, they are misleading when infectious agents are involved because they cannot detect or quantify different viral types or subtypes or degree and quality of inflammation to identify specific forms of the immune response [70,71].

### 3.1. Endomyocardial Biopsy

EMB is the gold standard method for diagnosing acute or chronic inflammatory heart disease. A definitive diagnosis depends on EMB. A 2013 European Society of Cardiology (ESC) position paper recommended the characterization of cardiac inflammation and infection by immunohistochemistry and viral genomic analysis with quantitative PCR (real-time PCR and nested PCR with reverse transcription) [1]. The major advantage of performing EMB in patients with suspected viral myocarditis and cardiogenic shock (Class I; C recommendation) or no recovery of cardiac function over 3 months despite optimal heart failure therapy (class 2A; C recommendation) is to exclude cardiac viral persistence. Histological, immunohistological and molecular analysis of EMB is a prerequisite for accurate diagnosis and represents the cornerstones for personalized therapy decisions in viral myocarditis. The ESC guidelines call for viral diagnostics that include viral genomic analysis of EMB samples by quantitative PCR. Importantly, the persistence of the viral genome is associated with progressive LV dysfunction, as demonstrated in follow-up biopsies, whereas spontaneous viral clearance is associated with improvement in systolic function. Furthermore, there is a need to exclude the presence of viruses in the EMB by PCR analysis before starting immunosuppression in patients with clinically suspected acute myocarditis. The ESC Working Group on Myocardial and Pericardial Diseases state that “Immunosuppression should be started only after ruling out active infection on EMB by PCR” [1] (Figure 2).

The development of the Dallas criteria based on traditional histological staining was a first attempt to standardize diagnostic guidelines. However, there are legitimate concerns about the diagnostic accuracy and sensitivity of the Dallas criteria because of ‘samp-ling errors’, which may be due to the focal and transient nature of the inflammatory process [72]. The use of immunohistochemistry markedly improves the number of EMB-revealing diagnoses of myocarditis. The use of different monoclonal antibodies allows the exact characterization and quantification of cellular infiltrates and cell adhesion molecules, both of which are relevant for the prognosis [73,74] (Figure 3).

In the area of modern molecular diagnostics, such as digital PCR (dPCR), real-time PCR of the 7th generation (fast qPCR), as well as next-generation sequencing, metagenomics, transcriptomics and other omics, rare, emerging and underestimated pathogens, as well as associated biomarkers, could be identified with pinpoint accuracy. Multiplex assays or metagenomics can detect co-infections and rare infections, and whole-genome sequencing or deep sequencing can identify viral variants, such as diagnostic and escape mutants.

### 3.2. Liquid Biopsy

Circulating biomarkers (liquid biopsy), including microRNAs (miRNAs), are already used for monitoring after heart transplantation and have the potential to complement EMBs [75]. MiRNAs are non-coding, small RNAs of 20–24 nucleotides [76]. They regulate gene expression at the post-transcriptional level and play a significant role in cell death, differentiation, cell signaling and various disease states [77,78]. Accumulated evidence has demonstrated the interactions between miRNAs and viruses [79]. MiRNA responses diverged, depending on the susceptibility to myocarditis after viral infection. MicroRNA-155, -146b, and -21 were consistently and strongly upregulated during acute myocarditis in both humans and susceptible mice [80,81].

During our own findings, investigating the role of inflammation and virus in patients with heart failure, we screened 754 unique circulating miRNAs in the serum of 343 biopsy-proven patients and identified 7 differently expressed miRNAs [82]. Based on the expression levels of let-7f, miR-197, miR-223, miR-93, miR-379, miR-21 and miR-30a-5p it was possible to discriminate between patients with viral myocarditis, inflammatory cardiomyopathy and healthy donors with a specificity of over 95%. Therefore, miRNA-panel may provide an excellent diagnostic capability. However, further studies must follow for elevating routine use.

## 4. Viruses

### 4.1. Enteroviruses/Adenoviruses

Enteroviruses (EV) are small, single-stranded ribonucleic acid (RNA) viruses. In the period from 1975 to 1985, the EV Coxsackievirus B3 (CVB3) was the most common myocarditis-causing virus worldwide [83]. To study the pathogenesis of viral myocarditis, mouse models of CVB3 infection are commonly employed. EV enter cardiomyocytes by binding to the transmembrane coxsackievirus and adenovirus receptor (CAR) and internalization (Figure 1). After uncoating, EV-RNA is directly translated into viral polyprotein in the cytoplasm and cleaved by multifunctional viral proteinases. The internal ribosome entry site drives viral translation and impedes cellular translation. From the antisense RNA strand matrices, EV-RNA polymerase transcribes the viral genome, which is packaged into viral particles. Especially the susceptible mouse strains A/J (H-2a) and Balb/c (H-2d) develop acute myocarditis post-infection. The disease is characterized by massive inflammatory infiltrates and necrosis. Direct damage to the heart in the acute phase is a consequence of viral replication and impaired cellular translation, induction of apoptosis and oxidative stress, followed by cell lysis. In the subacute phase of CVB3 infection, there may be an imbalanced immune response and immune-mediated destruction of cardiac tissue or induction of autoimmune processes, as indicated by the systemic presence of anti-myosin autoantibodies. In the chronic phase, CVB3 may be eliminated or viral persistence may lead to DCM, characterized by cytoskeletal dysfunction and impaired contractility. Viral persistence has been described particularly for 5′ terminally-deleted CVB3 variants, which are also associated with modulation of the IFN-ß pathway [45,80,84,85,86].

Recently, genome-wide association studies have linked specific genetic loci and innate and acquired response pathways with host responses against infectious diseases [87]. These results were not confirmed in the study by Belkaya et al., in which genetic mutations in IFN-mediated immunity examined in human pluripotent stem cell-derived cardiomyocytes did not result in increased susceptibility to viral myocarditis, although the study revealed a possible link between myocarditis and defects in cardiac structural proteins [88].

The clinical significance of persistent enteroviral genomes in the myocardium has been demonstrated by higher mortality. Data reported by Frustaci et al. from a retrospective analysis of immunosuppression-treated patients with inflammatory cardiomyopathy indicate that the situation in patients with persistent viruses did not improve or even worsened under immunosuppression, whereas the condition in virus-negative patients improved significantly [89]. Antiviral therapy with IFN-β results in the elimination of the virus from the myocardium, consistent with the improvement in LVEF [44] (Figure 4). Therefore, a precise diagnosis based on an EMB regarding viral persistence is required before a treatment decision can be made.

### 4.2. Erythroparvovirus

After a viral shift occurred, human parvovirus B19 (B19V) is the current most frequently detected viral species in EMBs of patients with suspected myocarditis or unexplained heart failure in Europe [43]. Long-term follow-up studies of biopsy-proven viral myocarditis revealed an incidence of 56–73% of B19V genome detection [90]. However, there are several studies suggesting that B19V persistence is not causative for myocarditis or heart failure [91,92]. Therefore, the clinical relevance of B19V in cardiac injury has been debated controversially in the past [92,93]. It has now been shown for the first time that B19V transcriptional activity is clinically relevant for the prognosis in DCMi (Figure 5 and Figure 6). Since the majority of clinical data are restricted to B19V genome (DNA) detection, the role of viral replication in the pathology of myocarditis still goes unregarded. Thus, differentiation between latent (inactive) and transcriptional active (mRNA positive) viral infection seems to be a considerable prognostic factor for the assessment of B19V-dependent cardiac damage (Figure 3A).

B19V infection is usually acquired during childhood and manifests as *erythema infectiosum*, mainly accompanied by mild symptoms or even asymptomatic courses. Accordingly, infection with B19V is widespread in the adult population and reflects a seroprevalence of about 60% [93]. The relatively small genome of B19V (5.6 kb) encodes for two major proteins (NS1 (non-structural protein) and VP1/2 (capsid protein)) and two small accessory proteins (11 kDa and 7.5 kDa) of largely unknown function. Parvoviral NS1 facilitates viral replication and host-cell apoptosis by protein transactivation of viral and human genes and by the induction of cell cycle arrest and DNA damage response. In this regard, various molecular mechanisms have been suggested for NS1-induced apoptosis and cytotoxicity [94]. In comparison, VP1/2 has phospholipase A2 activity that promotes host-cellular inflammation and cell lysis [95]. Due to its narrow host cell tropism, productive infection with B19V is restricted to erythroid-lineage cells and endothelial cells [96,97]. Thus, cardiac complications during B19V infection are widely accepted to be a consequence of endothelial dysfunction. Cardiomyocytes are not infected by B19V instead, infection of heart-resident endothelial cells leads to impaired coronary microcirculation and secondary myocyte damage [98]. Thereby, endothelial infection leads to an altered inflammatory signaling, an increased level of apoptosis and an elevated number of apoptotic circulating endothelial microparticles [99]. Furthermore, impaired endothelial regeneration and microvascular density were linked to myocardial B19V infection. In vitro infection of circulating angiogenic cells demonstrates B19V-dependent cellular damage and dysfunctional endogenous vascular repair [100]. In agreement, chronic B19V infection was associated with an impaired endothelial regenerative capacity [101]. Aside from direct cytotoxic effects, B19V is involved in autoimmunological processes [102]. It has been linked to the development of multiple autoimmune disorders, to cross-reaction with human antigens and to the release and presentation of self-antigens to t lymphocytes [103]. Furthermore, the phospholipase activity of VP1/2 is suggested to trigger autoimmunity, and VP1/2 immunization was shown to induce DCM in mice [104]. Finally, helicase and nickase activity of the NS1 protein has been shown to trigger autoimmunity [103,105].

Cardiac persistence of B19V infection with transcriptional activity has been associated with an adverse clinical outcome. It is proposed that B19V persistence (confirmed by serial biopsies) is responsible for the progression from myocarditis to DCM [106]. Furthermore, the immunization of mice with recombinant VP1/2 protein led to LVEF deterioration and the manifestation of cardiac fibrosis [104]. However, the sole detection of B19V genomes and evaluation of viral copy numbers have been proven to lack clinical significance [107,108]. In contrast to latent infection, the expression of B19V viral mRNA and NS1 and VP1/2 proteins in the myocardium was shown to be of clinical relevance [46]. Retrospective analyses of EMBs from patients with unexplained heart failure revealed an incidence of 26.9% with active viral transcription [46] (Figure 5). A five-year follow-up study of B19V-positive patients demonstrates that B19V replicative activity is related to an adverse clinical outcome (Figure 6). Thus, recent findings support the significance of B19V replication intermediates as the basis of an advanced EMB diagnostics. However, additional studies are required to elucidate the exact role of viral transcripts in the development of myocarditis and to further prove the significance of these markers in the clinical course of the disease.

There are numerous indications for a contribution of B19V co-infections for the progression to myocardial injury. Thus, B19V-induced cardiac damage could be enhanced by co-infection with other viral species, such as HHV6, influenza or even SARS-CoV-2 [109,110,111,112]. For instance, a recent autopsy report from a 5-year-old girl revealed that co-infection of myocardial B19V with respiratory influenza virus can cause fulminant myocarditis and pneumonia [109]. Adenoviral co-infection of endothelial cells induces B19V promoter transactivation and augmentation of VP1/2 and NS1 proteins [98]. Furthermore, EMB analysis of 498 B19V positive patients with myocarditis and DCM revealed that co-infection with HHV6 significantly impairs the recovery of heart failure patients after primary infection [111]. Aside from the direct interaction of HHV6 and B19V viral gene products, additive effects on the immune system are suggested to be involved in the deterioration of cardiac function [113,114].

### 4.3. Herpesviridae

Human herpesviruses 6A and 6B (HHV6A/B) are possible pathogenetic causes of myocarditis [12,113]. The prevalence of chromosomally integrated HHV6 (ciHHV6) is approximately 0.8% of HHV6-positive EMBs. The identification of individuals with ciHHV6 is important because the complete HHV6 genome is present in every cell of the body, and permanent reactivation of this virus is suspected in all tissues. Sustained high viral loads of HHV6 genomes in blood cells or tissues confirm the presence of ciHHV6.

Epstein–Barr virus (EBV), human cytomegalovirus (HCMV) and varicella-zoster virus (VZV) infections among immunocompetent adolescents and adults are rare causes of myocarditis and DCM, and only a few reports describe pathological phenotypes in the myocardium [8,39,115,116,117,118,119]. In immunocompromised individuals, such as transplant recipients, latent EBV, VZV, and HCMV infection could be reactivated and associated with many diseases of the cardiovascular system, including myocarditis, transplant vasculopathy, hypertension, restenosis and atherosclerosis [14]. Therefore, these herpes virus infections should be considered in the differential diagnostics of immunosuppressed patients, such as transplant recipients. Molecular mechanisms of the pathogenesis of EBV, HCMV and VZV infections of the myocardium remain to be elucidated; however, pathological mechanisms are presumed similar to other cardiotropic virus infections with direct cytopathic effects and secondary immune-mediated damage or endothelial dysfunction.

EBV is a ubiquitous herpesvirus (human γ-herpesvirus 4; HHV4) and causes infectious mononucleosis (glandular fever, kissing disease) and post-transplant lymphoproliferative disorders (Burkitt lymphoma, hemophagocytic lymphohistiocytosis and Hodgkin’s lymphoma) [120]. Acute EBV-related myocarditis is rare (<1%); however, in immunocompetent patients, the risk of life-threatening complications should not be underestimated, possibly due to the reactivation of latent EBV infection [121]. Vaccines against EBV are currently under development.

HCMV (human β-herpesvirus 5; HHV5) is a widespread virus that causes lifelong latent infection. HCMV infection has a high prevalence in the general population, ranging between 30% and 90% according to age, geographical region, and socioeconomic status. HCMV infection in immunocompetent individuals is usually asymptomatic or with mild symptoms; however, the clinical manifestation can vary due to the immune status. Cardiac involvement of HCMV infection is relatively uncommon in immunocompetent patients and was observed only slightly more frequently in fatal myocarditis cases [16,17,122]. However, in immunosuppressed patients, such as transplant recipients, several HCMV-associated myocarditis cases have been reported affecting 15–30% of this high-risk population [123]. HCMV vaccines are in clinical trials and have not yet been approved by the U.S. Food and Drug Administration or European Medicines Agency [124].

VZV (human α-herpesvirus 3; HHV3) causes chickenpox (varicella), a disease most commonly affecting children and adolescence after initial infection. Shingles (herpes zoster) is caused in adults mainly by reactivation of the virus. Risk factors for reactivation of VZV include old age and immunosuppression. Only very few cases have been reported that VZV and even reactivation of latent VZV infection can lead to cardiac complications, such as viral myocarditis and DCM [15]. VZV infection is preventable through vaccination [125].

### 4.4. Hepatitis C Virus and Human Immunodeficiency Virus

Chronic hepatitis C virus (HCV) infection is responsible for liver disease but may also be associated with various extrahepatic manifestations, such as mixed cryoglobulinemia, glomerulonephritis, lichen planus, myositis and others. The extrahepatic manifestations are thought to be due to the lymphotropic nature of HCV with the accumulation of circulating immune complexes, activation of autoimmune responses and modulation of host immune response [126]. According to the World Health Organization (WHO) an estimated 58 million people worldwide have chronic hepatitis C virus infection [127], while a high prevalence of HCV infection with cardiac diseases, such as myocarditis and DCM, has been found in 6.3% to 37% of HCV-positive individuals [9,10,11,128,129]. The pathogenesis of HCV-associated myocarditis, DCM and other cardiomyopathies is poorly understood. However, indirect immune-mediated and inflammatory mechanisms are thought to play a critical role in HCV-associated heart diseases. Mononuclear cells have been identified as the primary target cells of HCV in extrahepatic manifestations (monocytes/CD68^+^ macrophages) [130], and these cells are responsible for modulating the host immune response [62]. Thus, TNF-α has been shown to be one of the major proinflammatory cytokines involved in myocardial inflammation by stimulating nitric oxide expression and calcium homeostasis [131]. Direct-acting antivirals are effective therapeutic options for chronic HCV infection, not only for the treatment of liver disease but also for extrahepatic manifestation, such as HCV-associated heart diseases.

Human immunodeficiency virus (HIV) infections are associated with an increased incidence of cardiac complications. However, the pathophysiology is multifactorial. The pathogenesis of HIV-associated cardiomyopathy involves direct viral infection, cytokine activity, myocarditis, active antiretroviral therapy side effects, immune system dysregulation and/or ischemia [132].

### 4.5. SARS-CoV-2

Cardiac injury, including myocarditis, has been observed as a major complication in patients with coronavirus infection. Already previously identified coronavirus strains, such as severe acute respiratory syndrome-coronavirus-1 (SARS-CoV-1) and middle eastern respiratory syndrome (MERS), have been linked to myocardial damage [133,134]. Along with it, considering the dimension of knowledge gain due to the present COVID-19 pandemic, direct myocardial involvement and the pathogenic role of SARS-CoV-2 in myocarditis are still debated.

Primary infection with SARS-CoV-2 can lead to viral pneumonia and acute respiratory distress syndrome (ARDS), leading to an increased risk of morbidity and mortality [135]. Moreover, SARS-CoV-2 infection is associated with cytokine storm and coagulation abnormalities, leading to thromboembolic events up to multiple organ failure [136,137,138]. Conspicuously, there is a strong correlation between cardiovascular diseases (CVD) and the seriousness of COVID-19. It is supposed that susceptibility and clinical causes are highly dependent on cardiovascular comorbidities and that CVD risk factors, such as hypertension and diabetes, are closely related to fatal outcomes in COVID-19 patients [135,139,140,141,142]. Whether SARS-CoV-2 infection directly induces cardiac injury is still unclear. Numerous hypotheses ranging from direct myocardial interference over systemic inflammation (i.e., cytokine storm) to cardiometabolic issues and arrhythmias have touched this issue [143]. Although a vast number of meta-analyses, including several thousand COVID-19 patients, have proven the cardiac involvement of SARS-CoV-2, direct causality of coronavirus induced myocarditis and/or viral cardiomyopathy is not given so far [144,145,146,147].

Retrospective analysis of clinical data from first-generation COVID-19 patients reports a suspicion of myocarditis and myocardial injury in 12.5% of all patients [148]. The first indication of coronavirus localization in the heart was provided by ultrastructural analysis of EMB from a 69-year-old patient positively tested for SARS-CoV-2 [149]. Direct evidence of myocardial SARS-CoV-2 localization was provided by the detection of SARS-CoV-2 genomes in EMBs of patients with suspected myocarditis or unexplained heart failure [150]. Histological examination of heart tissue from COVID-19 deaths confirmed lymphocytic myocarditis in 1 out of 14 COVID-19 victims [151]. Another post-mortem analysis of 39 coronavirus victims revealed an incidence of 61.5% patients with SARS-CoV-2 RNA detection in the heart [152]. Although immune cell infiltration was not affected by the presence of SARS-CoV-2 in these patients, the expression of proinflammatory cytokines was increased. However, EMB analysis of patients with evidence of myocardial coronavirus infection showed an increased infiltration of inflammatory cells [153]. In agreement, a further case report of a patient with fulminant myocarditis revealed positive myocardial detection of SARS-CoV-2 RNA accompanied by massive macrophages and CD8^+^ cytotoxic t lymphocytes infiltration. Furthermore, in vitro data show an increased secretion of CCL2 cytokine from SARS-CoV-2-infected cardiomyocytes, resulting in an enhanced recruitment of monocytes [154]. Importantly, SARS-CoV-2 might also play a role in the pathogenesis of autoimmune myocarditis. Co-infection of SARS-CoV-2 and B19V is furthermore suggested to trigger the immune cascade resulting in fulminant myocarditis [110].

Taken together, several indications point towards a coronaviral involvement in the development of myocarditis. However, the dimension of cardiac damage and inflammatory response is heterogeneous, and the total incidence of myocarditis in COVID-19 patients is low. Thus, further studies are needed to understand the mechanisms of SARS-CoV-2 induced myocardial inflammation.

#### COVID-19 Vaccine and Myocarditis

Since the introduction of COVID-19 vaccines, there are several case reports indicating a relationship between the vaccination and the occurrence of myocarditis [155]. These cases reveal that the majority of COVID-19 vaccine-related myocarditis occurs in young male individuals following the second dose of the mRNA vaccines (Pfizer-BioNTech and Moderna COVID-19 vaccines). Imaging findings [156], as well as EMB analysis [157], confirmed signs of myocarditis developing within 2 weeks after COVID-19 vaccination (Figure 3D). However, a direct causal relationship has not been proven yet, and clinical symptoms generally ease quickly without impairment of cardiac function [155]. COVID-19 vaccine-related myocarditis shows fast recovery in the absence of short-term complications, and the overall incidence is marginal. Nevertheless, myocarditis is an alarming side-effect of COVID-19 vaccination, which needs to be monitored carefully.

### 4.6. Other Rare Virus Infections

Recently, the flaviviruses Zika virus (ZIKV), dengue virus (DENV) and West Nile virus (WNV), as well as the togavirus Chikungunya virus (CHIKV) and the lyssavirus rabies virus, all of which are endemic to tropical and subtropical regions, such as Latin America, and are neglected tropical diseases (NTD), have been found to be associated with cardiac diseases, such as myocarditis, in addition to neurological disorders. However, only a few case reports exist showing the association of these NTD virus infections with myocarditis [21,22,23,24,25,34,35,40,41,42,158,159]. Similar observations with a suspicious association to myocarditis have been made for rubella, mumps and measles virus [56]. Whether these viruses can be causative for the development of viral myocarditis or are just incidental findings needs to be determined.

## 5. Treatment Options

Patients with acute/chronic myocarditis/DCMi and heart failure are treated symptomatically with optimal heart failure medication according to the guidelines for the treatment of heart failure [68].

Patients with acute/chronic myocarditis/DCMi may experience life-threatening arrhythmias at any stage of the disease, which can lead to sudden cardiac death [160,161,162]. The high prevalence of cardiac electrical conduction disorders in patients with myocarditis underscores the clinical need to identify patients at risk for cardiac arrhythmias. The risk of sudden cardiac death in patients with acute myocarditis does not necessarily depend on the severity of the inflammation and may persist even after the inflammation has subsided [163]. Post-inflammatory ventricular arrhythmias, due to scarring, may present as monomorphic ventricular tachycardia in patients with healed myocarditis and occur in regions of increased myocardial fibrosis. Although efficacy has not yet been tested, antiarrhythmic drugs are usually used in patients with ventricular arrhythmias and acute myocarditis. Implantable cardioverter–defibrillator implantation should be considered after the acute inflammation has resolved [164]. A clear recommendation on the time point of implantation does not yet exist.

The definitive differentiation between virus positivity or virus exclusion, and the evidence of inflammation, its characterization and intensity are a fundamental prerequisite for a specific therapeutic decision based on EMB analyses. Specific studies on the antiviral treatment of chronic viral myocarditis are few and far between. The first randomized trial on viral cardiomyopathy was the placebo-controlled phase II multicentre BICC-Trial (Betaferon In Chronic Viral Cardiomyopathy) [7]. Compared to placebo, in enterovirus and adenovirus infections, IFN-β-1b was leading to effective virus clearance in follow-up EMB after treatment (Table 1). This was associated with favorable effects on NYHA functional class, improvement in quality of life and global patient assessment.

New antiviral strategies against B19V infections are under investigation and include the nucleotide analogs telbivudine. A recent study demonstrated the benefit of nucleoside analog treatment in controlling B19V replication and reducing viral transcripts, as well as rapidly improving symptoms in patients with active B19V infection [18,19] (Figure 7). To further explore the direct and important clinical impact of our findings, the results should be evaluated in a large randomized, site-controlled clinical trial that will provide further insight into effective B19V treatment conditions.

The initial data also suggest that treatment with IFN-β-1b leads to the suppression of transcriptional activity in B19V infection, associated with an improvement in hemodynamic outcomes (unpublished data).

Treatment with acyclovir and ganciclovir for herpes virus infection could be considered, although their efficacy has not been directly studied in patients with myocarditis. Ganciclovir (nucleoside analog of 2′-deoxy-guanosine) is suggested as a treatment option for severe EBV infection [165] (Table 1). Elimination of the chromosomally-integrated HHV6 virus is not possible, but initial data indicate that the transcriptional activity of ciHHV6 may be reduced under treatment with ganciclovir [13] (Figure 8).

A number of antiviral drugs for the treatment of VZV infection are available, including the nucleoside analogs acyclovir, famciclovir and the prodrug valaciclovir.

Patients with HIV-associated, HCV-associated or influenza-associated myocarditis are treated with established antiviral drugs, including antiretroviral therapy for patients with HIV [26,166]. HCV-associated myocarditis patients were treated with combination therapies of ombitasvir, paritaprevir, ritonavir and dasabuvir (Table 1).

Several antiviral—not cardiac-specific—therapies are currently being investigated for patients with COVID-19, including strategies to prevent the virus from entering the host cell, protease inhibitors (lopinavir-ritonavir and darunavir), RNA polymerase inhibitors (remdesivir) and anti-cytokine agents (e.g., IL-6 receptor antagonists). To date, clinical trial results with antiviral treatments showed no significant effects on mortality rate, length of hospital stay or other outcomes. The role of inflammasome in the pathogenesis and treatment of myocarditis needs further investigation [167].

Myocardial inflammation that persists after viral elimination requires immunosuppressive treatment to prevent subsequent autoimmune-mediated myocardial damage. However, viral genomes must be excluded prior to immunosuppressive therapy. Retrospective analyses showed that in patients with viral evidence, the prognosis did not improve or even worsened after immunosuppressive therapy [89]. Treatment for these patients with post-viral autoimmunological myocarditis consists of corticosteroids, azathioprine or ciclosporin A, in addition to optimal heart failure medication [168,169]. However, the subsequent therapeutic approach varies between individual patients and should be based on the underlying EMB analysis and adapted to the individual clinical development for a personalized treatment strategy.

The multicenter, double-blind, placebo-controlled, randomized phase 3 RHAPSODY trial demonstrated the efficacy and safety of rilonacept, an interleukin-1α and β inhibitor, in chronic pericarditis [170]. This agent may also be a potential therapeutic option for post-viral inflammatory processes in the myocardium in the future.

Cardiomyocyte-targeted RNA interference (RNAi) has been studied to inhibit cardiotropic viruses, such as human CVB3 and human ADV, in cardiomyocytes [171,172]. In addition to direct antiviral approaches, the potential of RNAi to suppress pathogenic cardiac inflammation has also been investigated [173]. The successful clinical implementation of such therapeutic approaches will depend on clinically safe drug delivery systems in future perspectives.

Patients with fulminant myocarditis presented with profound cardiogenic shock often need immediate mechanical circulatory support presented with profound cardiogenic shock often need mechanical circulatory support. Veno-arterial extracorporeal membrane oxygenation (VA-ECMO) is an excellent form of support for these patients as a bridge to the decision [174]. The time to recovery after the introduction of mechanical circulatory support varies considerably, ranging from days to weeks. While the majority of patients recover, a minority of patients with viral myocarditis require long-term mechanical circulatory support, with some ultimately requiring orthotopic heart transplantation.

As myocarditis is an underlying cause of sudden cardiac death in young athletes, current recommendations state that after a case of myocarditis, abstinence from competitive sports lasting between 3 to 6 months is generally recommended. This, however, can be extended to up to 1 year, but recommendations are based on scarce evidence [175,176].

## 6. Conclusions

Although the pathogenesis of acute/chronic viral myocarditis/DCMi has been studied in animal models and supported by EMB data in humans, the pathophysiological process raises further questions. There may be a number of genetic factors in humans that confer a preference for or protection against cardiac muscle injury, as has been suggested in animal models. The alteration of genetically determined immune responses alters viral elimination in some strains of mice from the heart during the acute phases of infection. These strains undergo significant remodeling of the heart, leading to a chronic form of myocarditis. Identifying the molecular basis for differences in the susceptibility of individuals could, therefore, open up new therapeutic opportunities to prevent or treat viral myocarditis in the future.

A specific and causal treatment adapted to pathophysiological characterized disease forms leads to a significant clinical improvement and a better prognosis of the patients. To reach this clinical prerequisite, there is the need for advanced diagnostics and guidelines to optimize the management of this disease. Large, randomized, controlled trials are needed to determine their role in the treatment of virally-induced cardiomyopathies.

## Figures and Tables

**Figure 1 jcm-10-05240-f001:**
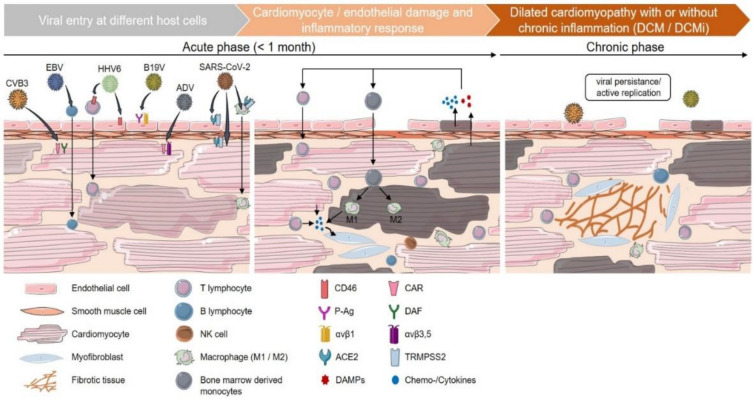
Phases of viral-mediated myocarditis. Cardiotropic viruses enter the myocardium via different host-cellular routes. Coxsackievirus B3 (CVB3) and adenovirus (ADV) directly target cardiomyocytes via the transmembrane coxsackievirus and adenovirus receptor (CAR). The decay-accelerating factor (DAF) serves as an additional CVB3 receptor, whereas integrins (αvβ3 and αvβ5) act as co-receptors for ADV internalization. Human herpesvirus 6 (HHV6) primarily targets CD4^+^ t lymphocytes and endothelial cells using CD46 as a cellular receptor. Epstein–Barr virus (EBV) enters cardiac tissue by infection of resting human b lymphocytes and subsequent infiltration into adjacent tissue. Parvovirus B19 (B19V) infects endothelial cells using erythrocyte P antigen (P-Ag) and integrin αvβ1 as a co-receptor. For SARS-CoV-2, several cardiac targets, such as cardiomyocytes, endothelial cells and circulating macrophages, are suggested. Its cellular entry depends on the expression of angiotensin-converting enzyme 2 (ACE2) and transmembrane protease serine subtype 2 (TMPRSS2) on the host-cell surface. Internalization of viral particles triggers a broad spectrum of host-cell responses and the activation of the innate immune system. Direct cardiomyocytolysis or apoptosis is induced by active viral replication and its transcription products (CVB3 and ADV). Indirectly, cardiomyocyte impairment can arise as a consequence of vascular endothelial dysfunction (B19V, HHV6). In addition, myocarditis is suggested to be induced by infected immune cells carrying viral genomes into the myocardium (HHV6, EBV). As a result of cardiac damage, inflammatory cytokines, as well as damage-associated molecular patterns (DAMPs), are released, triggering the infiltration of mononuclear cells, such as lymphocytes and monocytes, which differentiate into M1 or M2 macrophages depending on the inflammatory milieu. The further release of proinflammatory chemo and cytokines leads to the activation of heart-resident myofibroblasts and an increased generation of fibrous tissue. In the case of viral persistence, viral myocarditis can contribute to the chronic deterioration of cardiac function and the clinical presentation of dilated cardiomyopathy (DCM).

**Figure 2 jcm-10-05240-f002:**
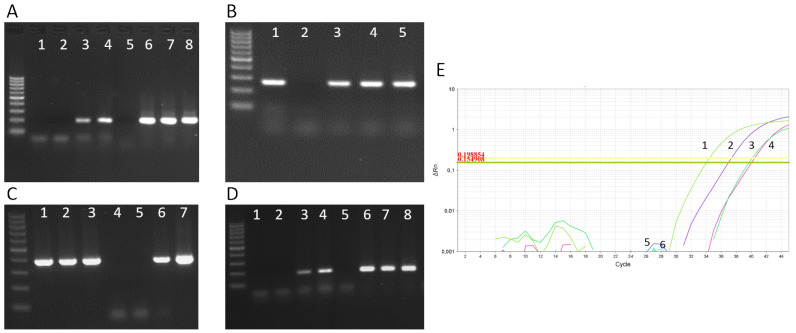
Gel electrophoresis blots (**A**–**D**) and real-time PCR amplification (**E**) of viral PCR amplicons of (**A**) three CVB3 positive samples (lanes 1 and 2 = ntc; lanes 3 and 4 = pc; lane 5 = negative patient sample; lanes 6–8 = CVB3 positive patient samples), (**B**) three EBV positive samples (lane 1 = pc; lane 2 = ntc; lanes 3–5 = EBV positive patient samples), (**C**) three HHV6 positive samples (lanes 1–3 = HHV6 positive patient samples; lane 4 negative patient sample; lane 5 = ntc; lanes 6 and 7 = pc), (**D**) three B19V positive patient samples (lanes 1 and 2 = ntc; lanes 3 and 4 = pc; lane 5 = negative patient sample; lanes 6–8 = B19V positive patient samples) and (**E**) one SARS-CoV-2 positive sample (1 = pc E-gene; 2 = pc Orf-gene; 3 = patient sample positive for SARS-CoV-2 E-gene; 4 = patient sample positive for SARS-CoV-2 Orf-gene; 5 = ntc E-gene; 6 = ntc Orf-gene). Abbreviations: ntc = no template control, pc = positive control.

**Figure 3 jcm-10-05240-f003:**
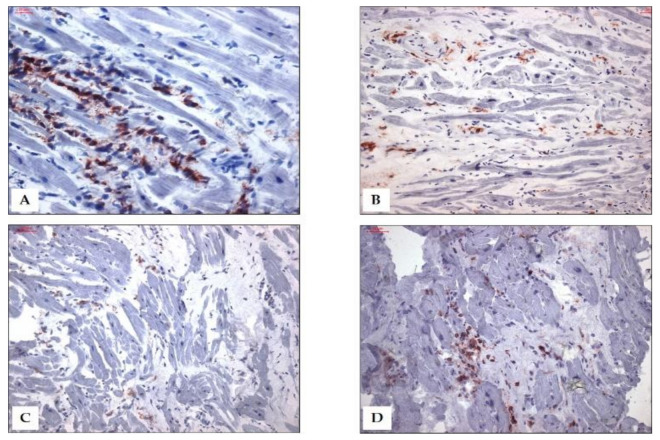
Representative immunohistological images. (**A**) Active myocarditis with B19V replicative activity. Massively increased CD3 positive t lymphocytes are stained in red/brown. Scale bar = 20 µm. (**B**) Inflammatory cardiomyopathy with positive detection of enterovirus genomes. Increased CD45R0^+^ T-memory cells are stained in brown. Scale bar = 50 µm. (**C**) EMB with positive detection of SARS-CoV-2 RNA. Increased cytotoxic perforin^+^ cells are stained in red/brown. Scale bar = 50 µm. (**D**) Staining of increased CD3^+^ t lymphocytes as a suspected consequence of COVID-19 vaccination. Scale bar = 50 µm.

**Figure 4 jcm-10-05240-f004:**
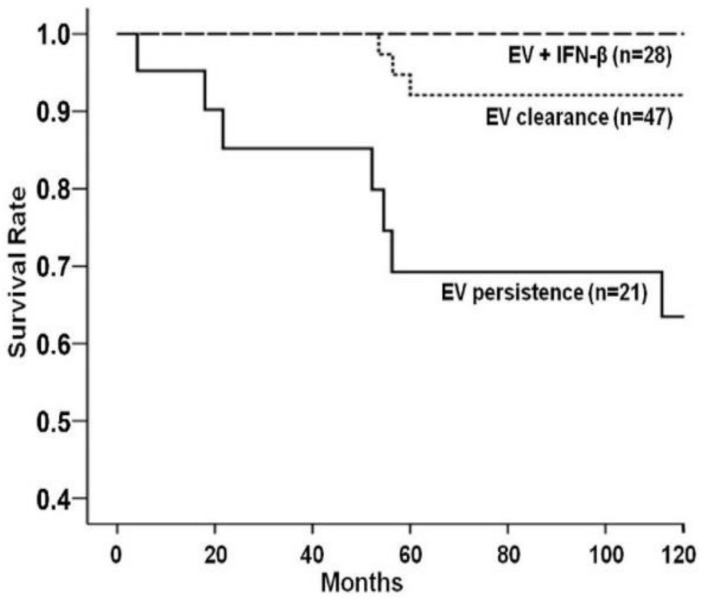
Mortality rate among patients positive for enterovirus (EV) infection: unadjusted survival according to virus analysis at follow-up. Spontaneous or interferon (IFN)-β-drug-induced enterovirus clearance was associated with a significantly reduced mortality rate in comparison to patients who had enterovirus persistence (*p* = 0.0005 by the log-rank test) [44].

**Figure 5 jcm-10-05240-f005:**
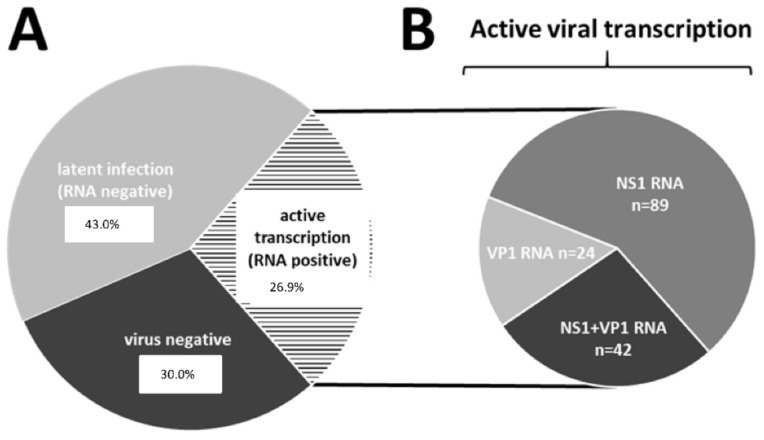
(**A**) Parvovirus B19 (B19V) genome detection and detection of viral transcription activity in EMBs of patients with unexplained heart failure (*n* = 576). (**B**) The group composition of EMBs with detectable active viral transcription (VP1/2-RNA-, NS1-RNA and VP1/2 and NS1-RNA positive samples) was shown in detail. Numbers represent the amount of EMBs. (**C**) The number of viral transcripts of NS1 compared to VP1/2. (**D**) Viral DNA loads in EMBs with active or latent infection. (**E**) Viral DNA load compared between EMBs with detectable VP1/2-RNA, NS1-RNA or NS1- and VP1/2-RNA expression (ANOVA *p* = 0.0427). Numbers above the bars represent *p*-values. Modified from [46].

**Figure 6 jcm-10-05240-f006:**
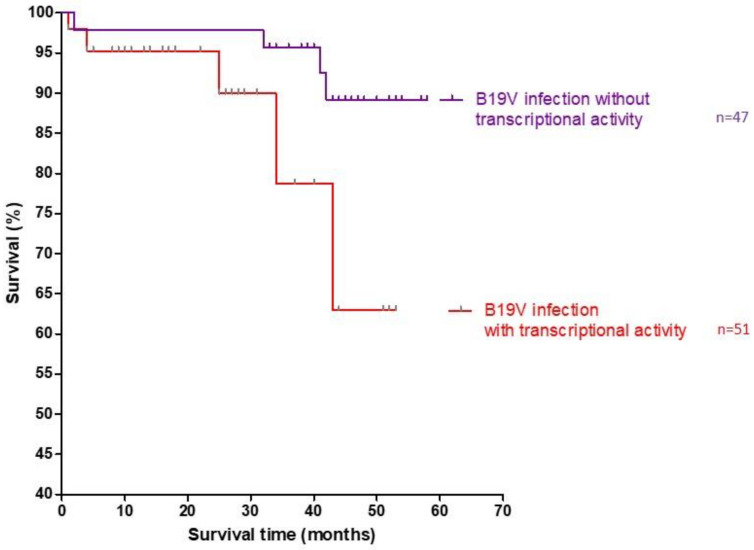
Kaplan–Meier plots. Natural all-cause mortality on long-term follow-up of B19V-positive patients with inflammatory cardiomyopathy in dependence of B19V transcriptional activity. The mortality rate was significantly higher in patients with transcriptional activity (*n* = 52) compared to those without transcriptional activity (*n* = 47) (*p* = 0.04 by the log-rank test).

**Figure 7 jcm-10-05240-f007:**
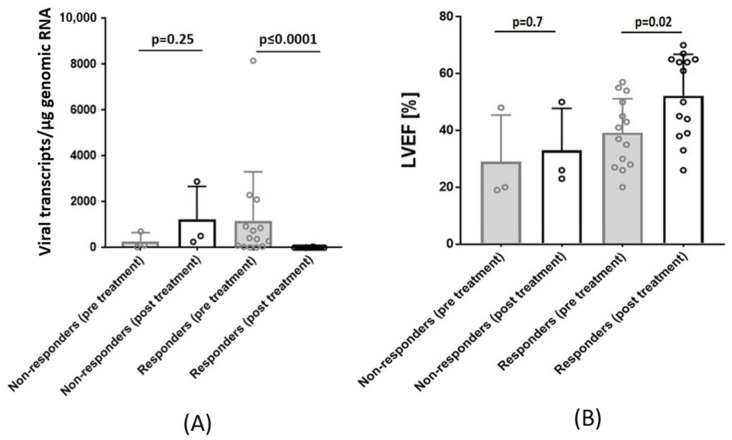
(**A**) Transcriptionally active cardiotropic B19V infection patients treated antivirally with the nucleoside analog telbivudine (LTD). Bar height indicates the mean value ± SD expression rate of viral transcripts/µg RNA in non-responders’ (*n* = 3) and responders’ group (*n* = 14) pre (baseline) and post (follow-up) LTD-treatment. Whereas responders significantly reduce viral replication intermediates upon treatment, non-responders show a non-significant increase of viral RNA. (**B**) LVEF of non-responders and responders pre and post-treatment with LTD. Bar height indicates LVEF (%) in non-responders’ and responders’ groups pre and post LTD-treatment. LVEF improvement was significantly improved in patients who reduced or lost the replicative viral intermediates (positive B19V RNA). LVEF is given as mean value and error bars represent SD [19].

**Figure 8 jcm-10-05240-f008:**
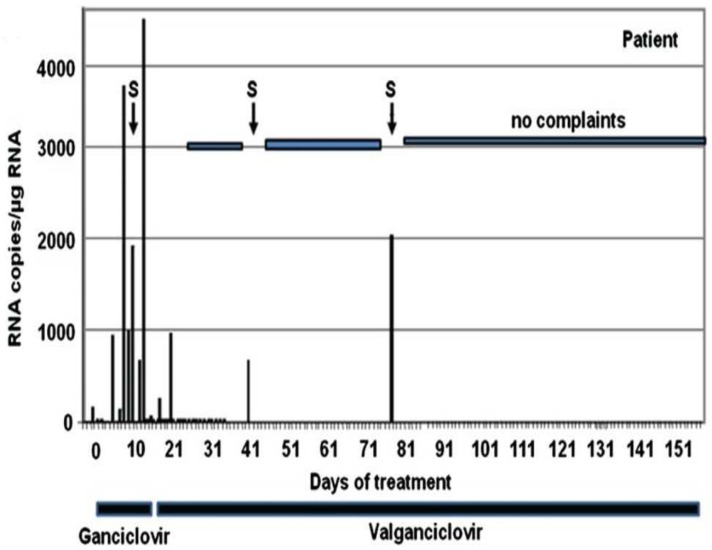
Example of a chromosomally integrated human herpesvirus 6B (ciHHV6B)-positive female patient with persisting high cardiac and systemic virus loads and cardiac involvement treated with antiviral gancyclovir. HHV6 RNA levels are shown as viral copy numbers per 1 µg of RNA. A symptomatic increase of mRNA (S) was noted between days 13 and 21 when i.v. ganciclovir was changed to oral administration. During short symptomatic phases (S) at days 41 and 75, again mild increases of mRNA were detected [13].

**Table 1 jcm-10-05240-t001:** Spectrum of viral infections in endomyocardial biopsy specimens with cardiac dysfunction in clinically suspected myocarditis and dilated cardiomyopathy.

Virus	Virus Family ^a^	Genome Organization	Target Cells in the Heart Muscle	Detection Frequency	Antiviral Therapy Option ^b^/Prevention	References
**Classical cardiotropic viruses**						
Adenovirus (ADV)	*Adenoviridae/human adenovirus*	linear dsDNA	cardiomyocytes	rare	β-interferon/n.v.a.	[6,7]
Enteroviruses (EV)/particularly Coxsackie viruses B (CVB), EV71 and echoviruses	*Picornaviridae*	linear (+) ssRNA	cardiomyocytes	moderate	β-interferon/n.v.a.	[7]
Epstein–Barr Virus (EBV)	*Herpesviridae/human γ-herpesvirus 4 (HHV-4)*	linear dsDNA	b lymphocytes, cardiomyocytes (?)	moderate	ganciclovir/n.v.a.	[8]
Hepatitis C virus (HCV)	*Flaviviridae*	linear (+) ssRNA	mononuclear cells/monocytes/CD68^+^-macrophages	infrequent	polymerase, proteinase, NS5A inhibitors, (e.g., sofosbuvir, simeprevir, daclastavir) etc./n.v.a.	[9,10,11]
Human herpesvirus-6 (HHV6)	*Herpesviridae/human β-herpesvirus*	linear dsDNA	CD4^+^ T lymphocytes, endothelial cells, T-cell tropism	frequent	ganciclovir, foscarnet, cidofovir/n.v.a.	[12,13]
Human cytomegalovirus (HCMV)	*Herpesviridae/human β-herpesvirus 5 (HHV-5)*	linear dsDNA	epithelial cells, endothelial cells, fibroblasts, smooth muscle cells	moderate	ganciclovir, valganciclovir/n.v.a.	[14,15,16,17]
Parvovirus B19 (B19V)	*Parvoviridae/erythroparvovirus*	linear (−) ssDNA	endothelial cells	frequent	[telbivudine; β-interferon] /n.v.a.	[18,19]
**Viruses with suspected association to myocarditis and DCM**						
Coronaviruses/particularly SARS-CoV-2	*Coronaviridae/severe acute respiratory syndrome coronavirus 2*	linear (+) ssRNA	cardiomyocytes, endothelial cells, macrophages	infrequent	n.d.a./licensed vaccines	[20]
Chikungunya virus (CHIKV)	*Togaviridae*	linear (+) ssRNA	epithelial cells, endothelial cells, primary fibroblasts, macrophages	infrequent	n.d.a./n.v.a.	[21,22,23]
Dengue virus (DENV)	*Flaviviridae*	linear (+) ssRNA	CD14+ monocytes	infrequent	n.d.a., symptomatic treatment/licensed vaccines	[24]
Herpes simplex virus 1 (HSV1)	*Herpesviridae/human α-herpesvirus 1*	linear dsDNA	epithelial cells, sensory neurons,	infrequent	acyclovir, valacyclovir, famciclovir/n.v.a.	[25]
Human immunodeficiency virus	*Retroviridae*	linear (+) ssRNA	dendritic cells, macrophages, osteoclasts	infrequent	antiretroviral therapy (ART)/n.v.a.	[26]
Hepatitis E virus (HEV)	*Hepeviridae/Orthohepevirus A*	linear (+) ssRNA	endometrial stromal cells, hepatocytes	infrequent	ribavirin, PEG-IFN-α/(vaccine licensed in China (HEV 239)	[27]
Influenza A and B viruses	*Orthomyxoviridae*	linear (+/−) ssRNA, segmented	human airway epithelial cells	infrequent	oseltamivir (Tamiflu^®^), zanamivir (Relenza^®^), peramivir (Rapivab^®^); baloxavir (Xofluza^®^)/licensed vaccine	[28,29]
Measles virus	*Paramyxoviridae*	linear (−) ssRNA	human airway epithelial cells	infrequent	n.d.a./licensed vaccine	[30]
Metapneumovirus	*Pneumoviridae*	linear (−) ssRNA		infrequent	n.d.a./n.v.a	[31]
Mumps virus	*Paramyxoviridae*	linear (−) ssRNA	neurotropic, CNS	infrequent	n.d.a./licensed vaccine	[32,33]
Rabies virus	*Rhabdoviridae*	linear (−) ssRNA	neurotropic, CNS	infrequent	post-exposure prophylaxis (PEP)/licensed vaccine	[34,35]
Respiratory syncytial virus (RSV)	*Pneumoviridae*	linear (−) ssRNA	human airway epithelial cells	infrequent	ribavirin/n.v.a.	[36,37]
Rubella virus	*Matonaviridae*	linear (+) ssRNA	human airway epithelial cells (?)	infrequent	n.d.a./licensed vaccine	[38]
Varicella-zoster virus (VZV)	*human α-herpesvirus 3 (HHV-3)*	linear dsDNA	sensory neurons	infrequent	acyclovir, famciclovir, valaciclovir/licensed vaccines	[39]
Zika virus (ZIKV)	*Flaviviridae*	linear (+) ssRNA	endometrial stromal cells	infrequent	n.d.a./n.v.a.	[40,41,42]

^a^ denotation according to the International Committee on Taxonomy of Viruses (ICTV). ^b^ for most antiviral therapies of patients with suspected myocarditis and dilated cardiomyopathy (DCM) there is currently no consensus regarding the indication of treatment. Detection rate in EMBs: infrequent = 0.1–1%, rare = 1–2%, moderate = 2–9%, frequent ≥ 10%. n.d.a. currently no specific antiviral drug available; n.v.a. currently no licensed vaccine available.

## Data Availability

The data presented in this study are available on request from the corresponding author.

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
