# Peer review of "Viral Myocarditis—From Pathophysiology to Treatment"

_jcm, 2021, doi:10.3390/jcm10225240_

Round 1
Reviewer 1 Report
The manuscript of Schultheiss and co-workers provides a clear and complete overview about viral myocarditis. Nevertheless, in my opinion, the following points deserve further clarification:
Major revisions
Line 40 – The authors state that “most cases of myocarditis are caused by infectious agents”. Actually, to the best of my knowledge, most cases of myocarditis observed in everyday clinical practice are clinically suspected and lack of any histological confirmation or PCR information about their etiology. The sentence should at least be explained or amended.
Line 449. The authors state that “cardiac involvement of coronaviruses is undisputed”. However, it should be explained that direct myocardial involvement and pathogenic role of SARS-Cov2 in myocarditis are still debated.
Line 253. Paragraph 4. Viruses. Association between HIV and myocarditis, though previously mentioned, is not treated.
Lines 591-592. It should be cleared that he role of autoinflammation and inflammasome in myocarditis needs to be further clarified. At present, we can only hypothesize a possible involvement of this pathogenic mechanism in some forms, as fulminant myocarditis, which is generally followed by spontaneous recovery.
Line 599. Fulminant myocarditis is a possible, common clinical presentation of lymphocytic, eosinophilic and giant cell myocarditis. Consequently, following stabilization after mechanical support, ensuing therapeutic approach is not the same in every patient, but it should be related to underlying histology and adapted to individual clinical evolution.
Minor revisions
Line 237 – Concerning miRNA and myocarditis, I suggest to include the following quotation:
A Novel Circulating MicroRNA for the Detection of Acute Myocarditis. Rafael Blanco-Domínguez, M.Sc., Raquel Sánchez-Díaz, Ph.D., Hortensia de la Fuente, M.D., Ph.D., et al. May 27, 2021N Engl J Med 2021; 384:2014-2027
Table 1, line five, square four, the sentence “CD4+ t lymphocytes, endothelial cells, t-cell tropism” should be changed as follow “CD4+ T lymphocytes, endothelial cells, T-cell tropism”.
Reviewer 2 Report
This is another excellently written review about myocarditis. Indeed it shares a lot with a recent review by Tschoepe et al. (https://doi.org/10.1038/s41569-020-00435-x) which was accordingly cited, but focuses on viral myocarditis. All in all, it is still a very good contribution in this field.
All in all, there are about 10 self references which seems adequate.
Figures are OK as well.
Here are some minor suggestions:
- On Page 4 lines 99-102. Please add reference.
- On Page 6. Here MRI and EMB are discussed. I completely agree with the indication of EMB. I also agree that MRI might be misleading in cases of infectious myocarditis. Indeed, the fact that myocarditis is infectious is only known after definite diagnosis with EMB. MRI is usually conducted before EMB. As a result, the question whether MRI should be performed in a clinical scenario in which myocarditis is suspected but the cause is uncertain should be discussed. I would suggest adding indications for MRI as it was done for EMB.
- I would suggest that the cessation of competitive sports after myocarditis should be at least shortly mentioned in e.g. section 5: Treatment Options.
Round 2
Reviewer 1 Report
Page 1, line 40
I would suggest to modify this sentence
"Most cases of myocarditis are probably caused by infectious agents, although often the pathogen cannot be detected after the onset of the disease",
with the following
"Many cases of myocarditis are likely caused or triggered by an infectious agent, although often the presence of a pathogen in myocardial tissue is not biopsy-proven".
Author Response
Comments: Page 1, line 40
I would suggest to modify this sentence
"Most cases of myocarditis are probably caused by infectious agents, although often the pathogen cannot be detected after the onset of the disease",
with the following
"Many cases of myocarditis are likely caused or triggered by an infectious agent, although often the presence of a pathogen in myocardial tissue is not biopsy-proven".
Reply: We are very grateful to the reviewer for his accurate correction. We have changed the sentence according to his suggestion. Thank you.
